# Controlling a Mecanum-Wheeled Robot with Multiple Swivel Axes Controlled by Three Commands

**DOI:** 10.3390/s25030709

**Published:** 2025-01-24

**Authors:** Yuto Nakagawa, Naoki Igo, Kiyoshi Hoshino

**Affiliations:** 1Degree Programs in Systems and Information Engineering, University of Tsukuba, Tsukuba 305-8577, Japan; nakagawa.yuto.qg@alumni.tsukuba.ac.jp; 2Faculty of Information Design, Tokyo Information Design Professional University, Tokyo 132-0034, Japan; 3School of Science and Technology, Meiji University, Kawasaki 214-8571, Japan; hoshino@esys.tsukuba.ac.jp

**Keywords:** Mecanum-wheeled robot, seven pivots, three commands, equation of the motion of the motor, PI control of the wheel rotation speed, control model

## Abstract

The Mecanum-wheeled robot has four special wheels. It can control four wheels independently and has seven turning axes. The robot can translate in all directions and travel in curves without changing its direction by means of the control commands for turning ratio, speed, and direction of travel. However, no model has been proposed that can accurately simulate the output of the actual machine for the three types of inputs, even when the characteristics of the motor and motor driver are unknown. In this study, we synthesized and simplified transfer functions and estimated the undetermined coefficients that minimize the sum of squared errors to construct a model of the robot that can output the position and posture equivalent to those of the actual robot for the input commands for turning ratio, speed, and the direction of travel. We modeled a Mecanum-wheeled robot using the proposed modeling method and parameter determination method and compared the outputs of the real robot to the step and ramp inputs. The results showed that the errors between the two outputs were very small and accurate enough to simulate AI learning, such as reinforcement learning, using the model of the robot.

## 1. Introduction

The Mecanum wheeled robot has four special wheels. It can control four wheels independently and has seven turning axes. Therefore, it can translate in all directions, turn, and travel in curves without changing direction by the means of three control commands: turning speed, wheel speed, and direction of the robot’s movement [1,2].

A driving simulator [3] that enables the user to experience a safe, realistic riding environment equivalent to that of a real car, and a robot that can perform the movements of the martial art of Tankendo are being developed using this robot as a moving mechanism [4,5]. In particular, the reference [5] reported that the relationship between the control commands and the corresponding output of the Mecanum-wheeled robot’s movements is nonlinear, which makes it difficult to predict the output, and that the robot cannot move instantaneously compared to a human being. Because of these characteristics, control methods for Mecanum-wheeled robots have been extensively studied [6,7,8,9,10,11,12,13,14,15,16,17]. For instance, the model of a Mecanum-wheeled robot is built using what is completely known about the properties of the motors and motor drivers [8], and has been used for research on disturbance elimination [9], the improvement of orbit-following performance [10], route following control [11], and control using SLAM [12]. However, it is uncertain whether it is possible to construct a quantitative model of a Mecanum-wheeled robot when unknown motors and drivers are used.

On the other hand, there are studies which use deep learning and reinforcement learning to represent the Tankendo motion with a Mecanum-wheeled robot [18,19]. In particular, when reinforcement learning is used, a model of the robot is constructed in the computer and learned using that model, and the accuracy of the model has a significant influence on the learning results. However, as mentioned above, the input–output relationships of robots are nonlinear and complex, and the models used in the references [9,10], for instance, are greatly simplified from the processing of Mecanum-wheeled robots, especially in that some nonlinear relationships are approximated linearly.

In this study, we aimed to construct a model part that can rigorously represent the nonlinear input–output relationship of a robot even when the properties of the motor and motor driver are unknown. Moreover, if the model is used for reinforcement learning, we aimed to achieve higher learning accuracy. In the modeling, we adopted the synthesis and simplification of transfer functions and the estimation of undetermined coefficients by the sum of squared errors. The final goal was to construct a robot model that could output the position and posture equivalent to those of a Mecanum-wheeled robot in response to input commands for the turning speed, wheel speed, and direction of the robot’s movement.

## 2. Specifications of the Mecanum Wheeled Robot

### 2.1. Main Body

Figure 1 shows the external appearance of the Mecanum wheeled robot that is the subject of the modeling in this study. A TDAM-STM manufactured by TOSA DENSHI, with a width of 977 [mm], depth of 775 [mm], height of 230 [mm], and weight of approximately 100 [kg], was used in this study.

### 2.2. Mecanum Wheel

#### 2.2.1. Configuration of the Mecanum Wheel

As shown in Figure 2, the robot was equipped with four Mecanum wheels with small barrel-shaped rollers at 45-degree angles to each axle. This generated a driving force in the direction of the wheel rotation and passive motion in the direction of the small barrel-shaped roller rotation. By controlling the four wheels independently, a variety of movements with seven pivots could be generated, as shown by the dots in Figure 3.

#### 2.2.2. Configuration of the Mecanum Wheeled Robot

The Mecanum-wheeled robot was given these inputs: turning speed $\omega \left(t\right)$ [deg/s], wheel speed $n\left(t\right)$ [rpm], and direction of the robot’s movement $\theta \left(t\right)$ [deg], which allowed the wheels to rotate. $n\left(t\right)$ was the wheel rotation speed when the robot moved straight forward, and the actual wheel rotation speed varied depending on $\omega \left(t\right)$ and $\theta \left(t\right)$. The input definitions are shown in Figure 4. Since Mecanum wheels were used, $\theta \left(t\right)$ could take values from 0 to 360 [deg]. And the maximum robot speed, V_max was multiplied by a function of $\theta \left(t\right)$ that had the range $1/\sqrt{2}<f\left\{\theta \left(t\right)\right\}<1$. The function $f\left\{\theta \left(t\right)\right\}$ was expressed as the length of the line segment extending from the origin at an angle $\theta \left(t\right)$ to the side of a square as shown in Figure 5. The following equation can be derived using the sine theorem.(1)$$\frac{f\{\theta (t)\}}{sin45}=\left\{\begin{array}{c} \begin{array}{c} \frac{1}{\sin\left\{180-45-\theta \left(t\right)\right\}}\;0\leq \theta (t)<90\\ \frac{1}{sin[180-45-\{180-\theta (t)\}]}\;90\leq \theta (t)<180 \end{array}\\ \begin{array}{c} \frac{1}{sin[180-45-\{\theta (t)-180\}]}\;180\leq \theta (t)<270\\ \frac{1}{sin[180-45-\{360-\theta (t)\}]}\;270\leq \theta (t)<360 \end{array} \end{array}\right.$$
By rearranging Equation (1), we can derive the equation of f{θ(t)}(2)$$f\{\theta \left(t\right)\}=\left\{\begin{array}{c} \begin{array}{c} \frac{1}{\sqrt{2}sin\{135-\theta (t)\}}\;0\leq \theta (t)<90\\ \frac{1}{\sqrt{2}sin\{\theta (t)-45\}}\;90\leq \theta (t)<180 \end{array}\\ \begin{array}{c} \frac{1}{\sqrt{2}sin\{315-\theta (t)\}}\;180\leq \theta (t)<270\\ \frac{1}{\sqrt{2}\sin\left\{\theta (t)-225\right\}}\;270\leq \theta (t)<360 \end{array} \end{array}\right.$$

## 3. Robot Modeling

### 3.1. Input/Output Specifications

Conventional models have not been able to represent the time lag between the inputs and outputs [5], but in this research, the nonlinear input–output relationship of the Mecanum-wheeled robot could be calculated by calculating the time variation of the four wheel speeds, including the time lag to the input, and then calculating the entire robot motion. The proposed model assumed operation in an ideal environment and modeled the kinematics and electrical characteristics of the robot and the controllers of the DC motors connected to the Mecanum wheels. Therefore, the friction forces, contact effects among the rollers in the Mecanum wheel, and the switching characteristics of the Mecanum wheel were not included in the model.

The inputs of the model were defined as the turning speed $\omega \left(t\right)$ [deg/s], wheel speed $n\left(t\right)$ [rpm], and the direction of robot movement $\theta \left(t\right)$ [deg]. These were the inputs defined by the robot manufacturer. And, the outputs were the displacements {X(t), Y(t)} [m] and posture change $\varphi \left(t\right)$ [deg] of the Mecanum-wheeled robot. $\varphi \left(t\right)$ was the angle between the robot’s frontal direction and the *X*-axis. The definition of outputs is shown in Figure 6. The block diagram of the entire robot model is shown in Figure 7.

### 3.2. Target Value to Target the Wheel Rotation Speed Conversion

The processing of the target value to target the wheel rotation speed conversion used the inverse kinematics of robot motion and wheel rotation speed presented by Taheri et al. [20].(3)$$\left[\begin{array}{c} \omega_{1}\\ \omega_{2}\\ \omega_{3}\\ \omega_{4} \end{array}\right]=\frac{1}{r}\left[\begin{array}{ccc} 1 & -1 & -\left(l_{x}+l_{y}\right)\\ 1 & 1 & \left(l_{x}+l_{y}\right)\\ 1 & 1 & -\left(l_{x}+l_{y}\right)\\ 1 & -1 & \left(l_{x}+l_{y}\right) \end{array}\right]\left[\begin{array}{c} v_{x}\\ v_{y}\\ \omega_{z} \end{array}\right]$$
where $\omega_{1}$–$\omega_{4}$ are the rotation speeds at each wheel, r is the radius of the Mecanum wheel, $l_{x}$ is half of the distance between the front wheels, $l_{y}$ is half of the distance between the front wheels and the rear wheels, $v_{x}$ and $v_{y}$ are the robot’s linear velocity, and $\omega_{z}$ is the robot’s turning speed. These definitions are shown in Figure 8.

Converting the units of $\omega \left(t\right)\;$ and n(t) to [rad/s], multiplying $\frac{1}{r}$ in Equation (3) by the rightmost matrix on the right side, and substituting $l_{x}+l_{y}=0.465\;$[m] and r=0.102 [m] according to the Mecanum-wheeled robot’s specifications, the determinant of the target value to the wheel rotation speed conversion part was derived.(4)$$\left[\begin{array}{c} \omega_{in1}\left(t\right)\\ \omega_{in2}\left(t\right)\\ \omega_{in3}\left(t\right)\\ \omega_{in4}\left(t\right) \end{array}\right]=\left[\begin{array}{ccc} 1 & -1 & -0.465\\ 1 & 1 & 0.465\\ 1 & 1 & -0.465\\ 1 & -1 & 0.465 \end{array}\right]\left[\begin{array}{c} n(t)sin\theta \left(t\right)\\ -n(t)cos\theta \left(t\right)\\ \frac{\omega \left(t\right)}{0.102} \end{array}\right]$$
where $\omega_{in1}\left(t\right)$–$\omega_{in4}\left(t\right)$ are the target rotational speed of each wheel.

### 3.3. Plant and Controller

#### 3.3.1. Transfer Function of the Plant

The plant is a DC motor in a Mecanum-wheeled robot. The transfer function of the plant section was specified from the differential equation of the DC motor. The differential equation of the DC motor could be obtained from Kirchhoff’s second law and the equation of the motion of rotation to obtain two equations.(5)$$L\frac{di\left(t\right)}{dt}+Ri\left(t\right)+K_{e}\omega_{out}\left(t\right)=e\left(t\right)$$(6)$$J\frac{d\omega_{out}\left(t\right)}{dt}+D\omega_{out}\left(t\right)=K_{t}i\left(t\right)$$
where $i\left(t\right)$ is the current flowing in the motor; $\omega_{out}\left(t\right)$ is the motor rotation speed, which is the model output; $e\left(t\right)$ is the input voltage to the motor; L is the inductance, R is the resistance; $K_{e}$ is the back EMF constant; $K_{t}$ is the torque constant; J is the moment of inertia; and D is the kinematic viscosity coefficient, which are motor specific values.

The Laplace transform of Equations (5) and (6) yields the following two equations.(7)$$sLI\left(s\right)+RI\left(s\right)+K_{e}\Omega_{out}\left(s\right)=E\left(s\right)$$(8)$$sJ\Omega_{out}\left(s\right)+D\Omega_{out}\left(s\right)=K_{t}I\left(s\right)$$

By substituting *I*(*s*), obtained from Equation (8), into Equation (7), the transfer function *P*(*s*) of the plant could be obtained.(9)$$\frac{\Omega_{out}\left(s\right)}{E\left(s\right)}=P\left(s\right)=\frac{K_{t}}{LJs^{2}+\left(LD+JR\right)s+\left(RD+K_{e}K_{t}\right)}$$

Equation (9) is a quadratic transfer function, but in practice, the units of parameter L  are [μF], and L≪1. To reduce the undetermined coefficient term, Equation (9) could be rewritten in the first-order form by approximating the parameter L as 0.(10)$$P\left(s\right)=\frac{\frac{K_{t}}{JR}}{s+\frac{RD+K_{e}K_{t}}{JR}}$$

#### 3.3.2. Transfer Function of the Controller

The controller in this study was assumed to be a PID controller because the word “PID control” is found in the specifications of the motor BLV620KM20S-1 used in the actual robot. However, the controller in this study was assumed to be a PI controller with Equation (11) to reduce the number of orders.(11)$$e\left(t\right)=Kp\left(\omega_{in}\left(t\right)-\omega_{out}\left(t\right)\right)+Ki\int_{0}^{t}\left(\omega_{in}\left(\tau \right)-\omega_{out}\left(\tau \right)\right)d\tau$$
where $e\left(t\right)$ is the output of the controller, the voltage applied to the controlled object; $\omega_{in}\left(t\right)$ is the input to the controller, the target rotational speed; $\omega_{out}\left(t\right)$ is the feedback of the rotational speed of the plant; Kp is the proportional gain; and Ki is the integral gain.

#### 3.3.3. Transfer Function of the Entire Model

Equation (12) is the Laplace transform of Equation (11).(12)$$E\left(s\right)=Kp\left(\Omega_{in}\left(s\right)-\Omega_{out}\left(s\right)\right)+\frac{1}{s}Ki\left(\Omega_{in}\left(s\right)-\Omega_{out}\left(s\right)\right)$$
Eliminating E(s) by Equations (10) and (12) and rearranging for $\Omega_{in}\left(s\right)$ and $\Omega_{out}\left(s\right)$, the motor model could finally be represented by a single transfer function *C*(*s*)*P*(*s*).(13)$$C\left(s\right)P\left(s\right)=\frac{\Omega_{out}\left(s\right)}{\Omega_{in}\left(s\right)}=\frac{\frac{K_{t}K_{p}}{JR}s+\frac{K_{t}K_{i}}{JR}}{s^{2}+\left(\frac{RD+K_{e}K_{t}+K_{t}K_{p}}{JR}\right)s+\frac{K_{t}K_{i}}{JR}}$$
Equation (13) consists of a combination of a first-order advanced transfer function and a quadratic transfer function, and the constant terms $\frac{K_{t}K_{i}}{JR}$ in the denominator and numerator are common, so the number of undetermined coefficients can be reduced by one.

### 3.4. Wheel Rotation Speed to Robot Motion Conversion

The inputs to the wheel rotation speed to robot motion conversion were the outputs $\omega_{out1}$–$\omega_{out4}\;$ from the motor model, and the outputs were the coordinates $X\left(t\right)$ and $Y\left(t\right)$ of the robot’s center of gravity and the posture $\varphi \left(t\right)$.

The sequential kinematics of wheel rotation and robot motion has also been shown by Taheri et al. [20].(14)$$\left[\begin{array}{c} v_{x}\\ v_{y}\\ \omega_{z} \end{array}\right]=\frac{r}{4}\left[\begin{array}{c} 1\\ -1\\ \frac{-1}{\left(l_{x}+l_{y}\right)} \end{array}\begin{array}{c} 1\\ 1\\ \frac{1}{\left(l_{x}+l_{y}\right)} \end{array}\begin{array}{c} 1\\ 1\\ \frac{-1}{\left(l_{x}+l_{y}\right)} \end{array}\begin{array}{c} 1\\ -1\\ \frac{1}{\left(l_{x}+l_{y}\right)} \end{array}\right]\left[\begin{array}{c} \omega_{1}\\ \omega_{2}\\ \omega_{3}\\ \omega_{4} \end{array}\right]$$

Substituting the corresponding values in Equation (15), the following equation is derived.(15)$$\left[\begin{array}{c} v_{x}(t)\\ v_{y}(t)\\ \omega_{z}(t) \end{array}\right]=\frac{0.102}{4}\left[\begin{array}{c} 1\\ -1\\ \frac{-1}{0.465} \end{array}\begin{array}{c} 1\\ 1\\ \frac{1}{0.465} \end{array}\begin{array}{c} 1\\ 1\\ \frac{-1}{0.465} \end{array}\begin{array}{c} 1\\ -1\\ \frac{1}{0.465} \end{array}\right]\left[\begin{array}{c} \omega_{out1}(t)\\ \omega_{out2}(t)\\ \omega_{out3}(t)\\ \omega_{out4}(t) \end{array}\right]$$

By transformation of the robot coordinate system and the global coordinate system, the *X*-axis velocity $v_{X}\left(t\right)$, *Y*-axis velocity $v_{Y}\left(t\right)$, and turn speed $\omega_{z}\left(t\right)$ in the global coordinate system was calculated by the following equation.(16)$$\left[\begin{array}{c} v_{X}\left(t\right)\\ v_{Y}\left(t\right)\\ \omega_{z}\left(t\right) \end{array}\right]=\left[\begin{array}{ccc} cos\varphi^{*} & -sin\varphi^{*} & 0\\ sin\varphi^{*} & cos\varphi^{*} & 0\\ 0 & 0 & 1 \end{array}\right]\left[\begin{array}{c} v_{x}\left(t\right)\\ v_{y}\left(t\right)\\ \omega_{z}\left(t\right) \end{array}\right]$$
where $\varphi^{*}$ is the angle between the robot coordinate system and the global coordinate system and is equal to the robot’s posture $\varphi \left(t\right)$, calculated before the temporal tick.

Outputs $X\left(t\right)$, $Y\left(t\right),\;\varphi \left(t\right)$ were calculated by taking the integral of $v_{X}\left(t\right)$, $v_{Y}\left(t\right)$, $\omega_{z}\left(t\right)$, and adding them to the initial values $X_{0}$, $Y_{0},\;\varphi_{0}$.(17)$$\left[\begin{array}{c} X\left(t\right)\\ Y\left(t\right)\\ \varphi \left(t\right) \end{array}\right]=\left[\begin{array}{c} X_{0}\\ Y_{0}\\ \varphi_{0} \end{array}\right]+\int_{0}^{t}\left[\begin{array}{c} v_{X}(\tau )\\ v_{Y}(\tau )\\ \omega_{z}\left(\tau \right) \end{array}\right]d\tau$$

Consequently, the wheel rotation speed to robot motion conversion was described by the following equation derived from Equations (15)–(17).(18)$$\left[\begin{array}{c} X\left(t\right)\\ Y\left(t\right)\\ \varphi \left(t\right) \end{array}\right]=\left[\begin{array}{c} X_{0}\\ Y_{0}\\ \varphi_{0} \end{array}\right]+\frac{0.102}{4}\int_{0}^{t}\left[\begin{array}{ccc} cos\varphi^{*} & -sin\varphi^{*} & 0\\ sin\varphi^{*} & cos\varphi^{*} & 0\\ 0 & 0 & 1 \end{array}\right]\left[\begin{array}{c} 1\\ -1\\ \frac{-1}{0.465} \end{array}\begin{array}{c} 1\\ 1\\ \frac{1}{0.465} \end{array}\begin{array}{c} 1\\ 1\\ \frac{-1}{0.465} \end{array}\begin{array}{c} 1\\ -1\\ \frac{1}{0.465} \end{array}\right]\left[\begin{array}{c} \omega_{out1}(t)\\ \omega_{out2}(t)\\ \omega_{out3}(t)\\ \omega_{out4}(t) \end{array}\right]dt$$

### 3.5. Undetermined Coefficient Estimation

#### 3.5.1. Undetermined Coefficient Setting

To operationalize the model, it was necessary to estimate the undetermined coefficients of the motor model transfer function described by Equation (13). Equation (13) is represented by the DC motor-specific constants L, R, $K_{e}$, $K_{t}$, J and D  and the PI gains $\;K_{p}$, $K_{i}$. However, it was difficult to obtain the motor-specific constants by measurement because the DC motor was already integrated in the robot. Therefore, letting the parameters $A=\frac{\;K_{t}K_{p}}{JR}$, $B=\frac{RD+K_{e}K_{t}}{JR}$, $C=\frac{K_{t}K_{i}}{JR}$, and rewriting Equation (13) as the following equation, we let the coefficient estimation be the problem of finding the undetermined coefficients A, B, and C.(19)$$C\left(s\right)P\left(s\right)=\frac{As+C}{s^{2}+(A+B)s+C}$$

Next, we considered the zero-points and poles in Equation (19). The zero-point was $s=-\frac{C}{A}$, since As+C = 0 is satisfied. The pole was the solution of the denominator $s^{2}+(A+B)s+C$ = 0 and could be represented by the following equation.(20)$$s=\frac{-(A+B)\pm \sqrt{{\left(A+B\right)}^{2}-4C}}{2}$$

If the real part of s was negative, then the model was stable. Since the constant contained in parameter *C* was positive, the following equation was obtained.(21)$$\sqrt{{\left(A+B\right)}^{2}-4C}<A+B$$

The real part of the poles was negative regardless of the parameters *A*, *B*, and *C*, and so the model was stable.

#### 3.5.2. Estimation of the Three Undetermined Coefficients

First, to obtain the undetermined parameters for A, B, and C, the parameter B, which was close to the characteristics of the motor, was set to 15.5. This value was calculated from the value of the motor CPH62, whose performance is similar to that of the DC motor BLV620KM20S-1 used in the Mecanum-wheeled robot and whose parameters are publicly available. A comparison of the performance of each motor is shown in Table 1. Since a wheel was actually attached to the motor rotation axis and the moment of inertia of the wheel was given as 8.61 $\times {\;10}^{-4}$ [kg·m^2^] from the robot specifications, a value of inertia moment J was used as shown in the following equation.(22)$$J=8.61\times 10^{-4}+1.97\times 10^{-4}$$

Second, the parameters A and C were varied in the range of 0 to 2000 with an increment of 1 to find the values of the parameters A and C that minimized the sum of the squares of the error between the model and the actual step response. The range of 0 to 2000 corresponded to the range of approximately $0\leq K_{p}$,$\;K_{i}\leq 10$ in the PI gains. The increment was 1, corresponding to the increment of 0.005 in the PI gain, which was sufficiently detailed. The input was a one second step input where $\omega \left(t\right)$ and θ$\left(t\right)$ were constant at 0 [deg/s] and 90 [deg], and only $n\left(t\right)$ changed from 0 to 60 [rpm] at time 0. The reason for this was to match the real and model outputs in translation only. The reason n(t) = 60 [rpm] was because the maximum value was approximately 120–130 [rpm], and 60 [rpm] was an intermediate value.

Third, to fix the coefficients A and C, B was varied again in increments of 0.1 from 5.5 to 25.5, and recalculated to provide the coefficient $B^{*}$ that minimized the sum of squares of the errors.

Fourth, to fix $B^{*}$ and find the combination of A and C that minimized the error through the coefficients $A^{*}$ and $C^{*}$, the range of values for A and C was again set to be from 0 to 2000. In order to match the number of significant figures here, four decimal places were rounded to the nearest whole number.

Since the transfer functions for the four motor models were identical, the coefficients obtained could be applied to all motor model parts.

As a result, the following combinations were obtained: $A^{*}$ = 14.7, $B^{*}$ = 16.1, and $C^{*}$ = 303. A comparison of the model outputs when these parameters were substituted and the wheel rotation speed of the Mecanum-wheeled robot is shown in Figure 9.

## 4. Experiments Comparing the Mecanum-Wheeled Robot and Model Outputs

### 4.1. Wheel Rotation Speed Comparison Between the Model and Mecanum-Wheeled Robot

To evaluate the model, two evaluation experiments were conducted. The number of rotations output from the robot was measured with the robot’s wheels not in contact with the ground. Markers were attached to the wheels, the rotation of the wheels was photographed by a camera, and the number of rotations was calculated by image processing. The first was a wheel rotation output comparison between the model and the Mecanum wheeled robot for step inputs. Figure 10 shows the wheel rotation speed comparison for one second step inputs where $\omega \left(t\right)$ and $\theta \left(t\right)$ were constant at 0 [deg/s] and 90 [deg], and only $n\left(t\right)$ changed from 0 to 30, 60, and 90 [rpm] at time 0. Figure 11 shows the wheel rotation speed comparison for one second step inputs where $\omega \left(t\right)$ and $\theta \left(t\right)$ were constant at 0 [deg/s] and 90 [deg], and only $n\left(t\right)$ changed from 0 to 100 [rpm] at time 0. Figure 12 is a wheel rotation speed comparison for a ramp input where $\omega \left(t\right)$ and $\theta \left(t\right)$ were constant at 0 [deg/s] and 90 [deg], and only $n\left(t\right)$ increased from 0 at time 0 with a slope of 5 [rpm/s].

The experimental results with the step inputs as shown in Figure 10 indicated that the proposed model was able to simulate the Mecanum-wheeled robot’s output with little error between the robot’s output and the proposed model’s output. As the results in Figure 11 show, there was a delay in the rise time of the Mecanum-wheeled robot compared to the proposed model when 100 [rpm] was given as the input $n\left(t\right)$. These were thought to be due to the decrease in torque associated with the increase in rotational speed.

When a ramp input of 5 [rpm/s] was inputted, as shown in Figure 12, the wheel rotation speed did not increase when the input value exceeded 130 [rpm]. Therefore, the hardware wheel rotation speed limit of this Mecanum-wheeled robot was in the neighborhood of 130 [rpm]. However, the model was able to reproduce the output of the actual machine well in the range where $n\left(t\right)$ was less than 100 [rpm].

### 4.2. Whole Outputs Comparison Between the Robot and Model

Second, the center-of-gravity position and posture outputs of the model and the Mecanum-wheeled robot were compared when the step and ramp inputs were applied. Figure 13 compares the model and actual output for a step input where $\omega \left(t\right)$ and $\theta \left(t\right)$ were constant at 0 [deg/s] and 90 [deg], and $n\left(t\right)$ changed from 0 to 120 [rpm] at time 0. Figure 14 compares the model and actual output for a step input where $n\left(t\right)$ and $\theta \left(t\right)$ were constant at 0 [rpm] and 90 [deg], and $\omega \left(t\right)$ changed from 0 to 60 [deg/s] at time 0. Figure 15 compares the model and actual output for a ramp input with $n\left(t\right)$ = 5t [rpm] while $\omega \left(t\right)$ and $\theta \left(t\right)$ were constant at 0 [deg/s] and 90 [deg].

The experimental results in Figure 13, Figure 14 and Figure 15 showed that there were few errors between the Mecanum-wheeled robot’s output and the output of the proposed model, and the output of the proposed model was able to simulate the robot ’s output. In particular, Figure 13 shows that good output was obtained even when $n\left(t\right)$ = 120 [rpm], and Figure 14 shows that the model could also represent the Mecanum-wheeled robot’s turning. The experimental results suggested the validity of the model and the coefficient estimation.

## 5. Conclusions

The Mecanum-wheeled robot has four special wheels. It can control four wheels independently and has seven turning axes. Therefore, it can translate in all directions, turn, and travel in curves without changing direction by means of three control commands: turning speed, wheel speed, and the direction of the robot’s movement. In this study, we aimed to construct a model part that could rigorously represent the nonlinear input–output relationship of a robot even when the properties of the motor and motor driver were unknown. In the modeling, we adopted synthesis and simplification of the transfer functions and the estimation of undetermined coefficients by the sum of squared errors. The final goal was to construct a robot model that could output the position and posture equivalent to those of the Mecanum-wheeled robot in response to input commands for the turning speed, wheel speed, and direction of the robot’s movement.

Specifically, the Mecanum-wheeled robot was modeled as a system consisting of three types of blocks: a converter from the input commands to the robot body to the rotational speed commands of the four motors, four sets of controllers (i.e., motor drivers) and a control target (i.e., motor), and an output converter from each motor rotational speed command output from the four sets of controllers and the control target to the robot body. For the control object, the transfer function was obtained by Laplace-transforming Kirchhoff’s second law and the equation of the motion of rotation, respectively. By setting the inductance term to zero, the system was assumed to be linear. The controller was a PI control. The transfer functions of the controller and the controlled object were combined and simplified to be a combination of a first-order advance system and a quadratic system. To determine the values of the three undetermined coefficients obtained, the value that minimized the sum of the squared errors of the Mecanum-wheeled robot and model outputs was determined.

Two experiments were conducted to evaluate the validity of the model and the coefficient values. The first compared the output of a single motor (i.e., wheel) of the Mecanum-wheeled robot with the output of the motor drive and the modeled part of the motor when the robot was given step and ramped commands with a known rotation speed of a single motor in the robot. The second was a comparison of the outputs of the Mecanum-wheeled robot and the whole model when step and ramp inputs were given.

As a result, in the first experiment, a delay in the rise of the rotation speed was observed near the maximum command value in the Mecanum-wheeled robot, but this was not observed in the ideal model output. However, there was good agreement between the Mecanum-wheeled robot and the model outputs except near the maximum value. In the second experiment, the difference between the output of the Mecanum-wheeled robot and that of the model was extremely small. In addition, the model was able to represent the behavior of the Mecanum-wheeled robot well even when given a command to rotate.

A number of the experimental results suggested the validity of the model and the validity of the coefficient estimation. This study proposed a method for determining parameters necessary for modeling a Mecanum-wheeled robot controlled by three input commands so that the robot could be modeled even when the characteristics of the motor and motor driver were not strictly known. The proposed parameter determination method can be applied to other robots whose motor and motor driver characteristics are not strictly known. In addition, various simulations can be performed using the realized model of the Mecanum-wheeled robot. For example, the realized robot model can be used to train robot control AI for a short sword training system using a Mecanum-wheeled robot. Therefore, this study can greatly contribute to the parameter determination method in robot modeling and to the realization of application systems using Mecanum-wheeled robots.

## Figures and Tables

**Figure 1 sensors-25-00709-f001:**
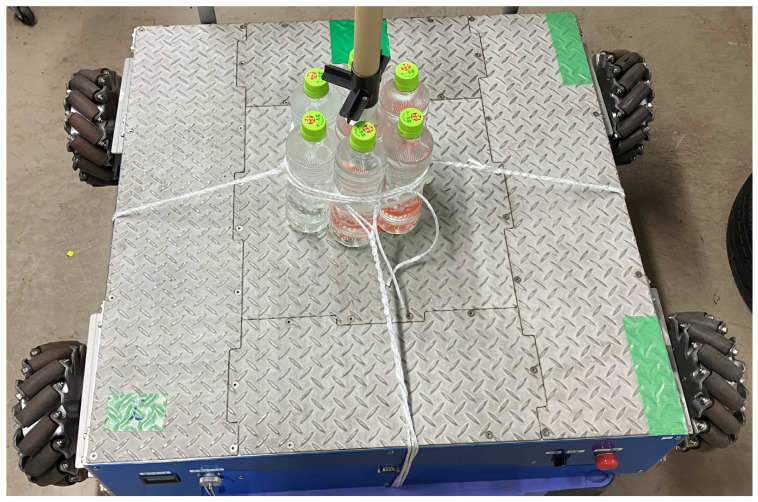
External appearance of the Mecanum-wheeled robot.

**Figure 2 sensors-25-00709-f002:**
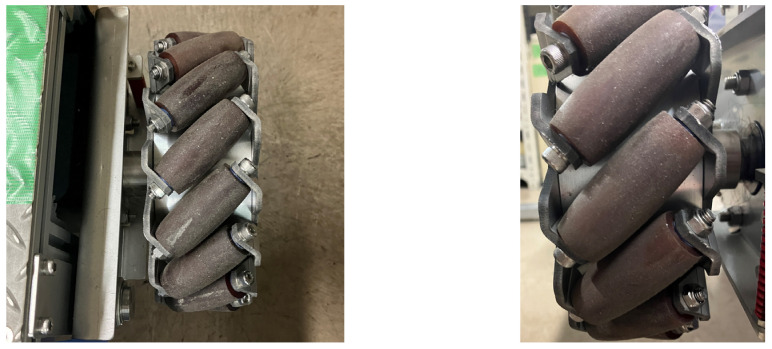
Barrel-shaped rollers at 45-degree angles to each axle.

**Figure 3 sensors-25-00709-f003:**
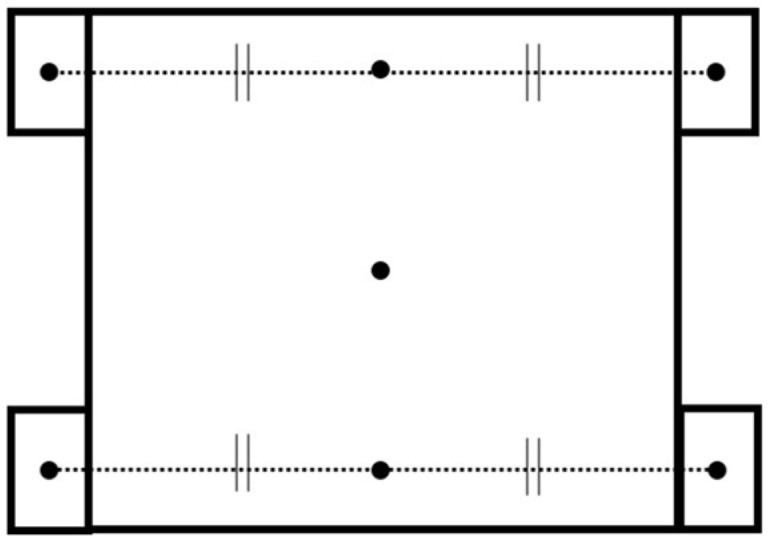
Seven pivots of the Mecanum-wheeled robot.

**Figure 4 sensors-25-00709-f004:**
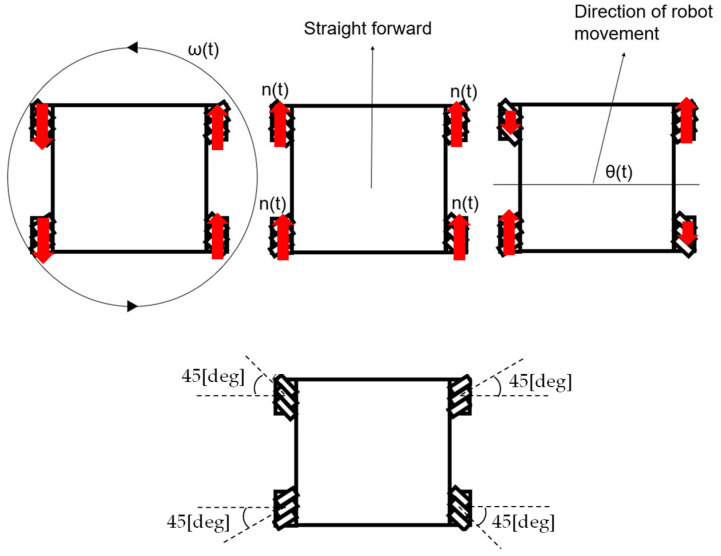
Definition of inputs.

**Figure 5 sensors-25-00709-f005:**
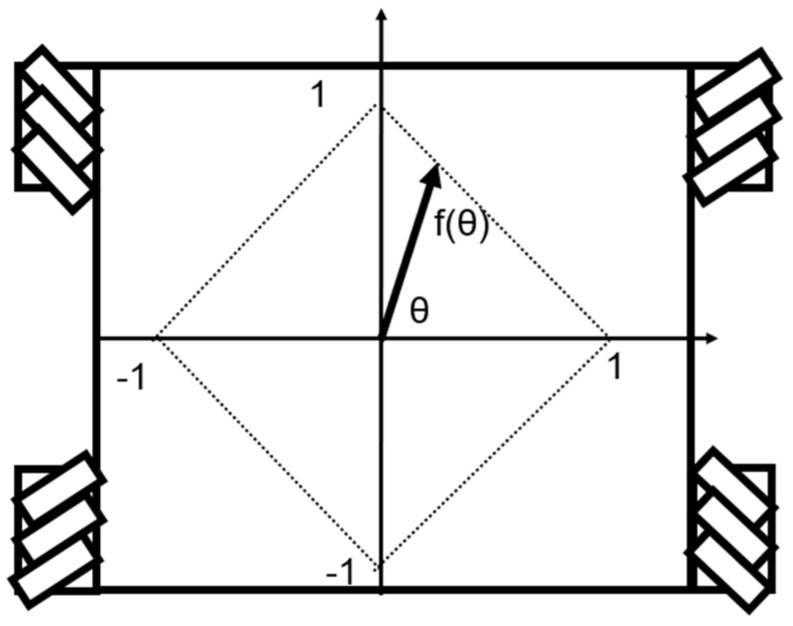
Definition of f(θ).

**Figure 6 sensors-25-00709-f006:**
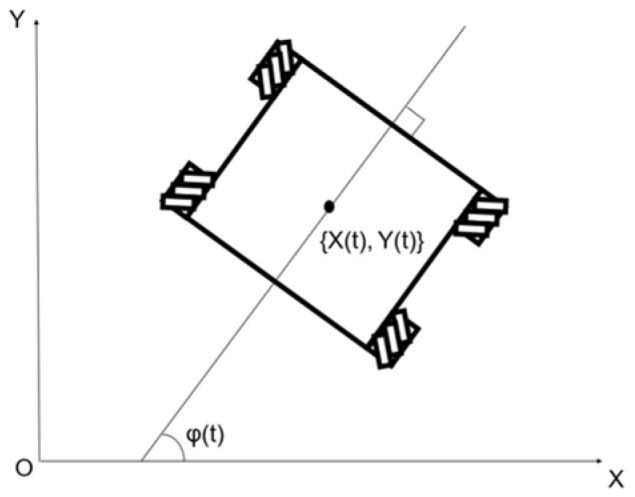
Definition of outputs.

**Figure 7 sensors-25-00709-f007:**
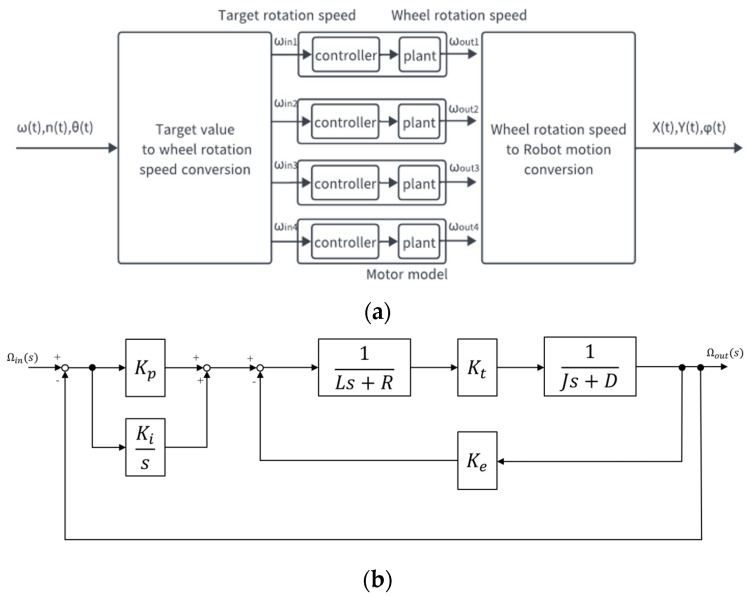
Block diagram of the robot model. (**a**) Block diagram of the entire robot model. (**b**) Block diagram of the controller and plant.

**Figure 8 sensors-25-00709-f008:**
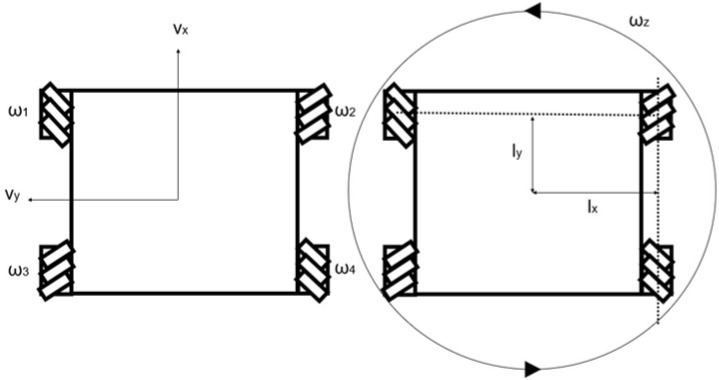
Definition of the robot’s parameters.

**Figure 9 sensors-25-00709-f009:**
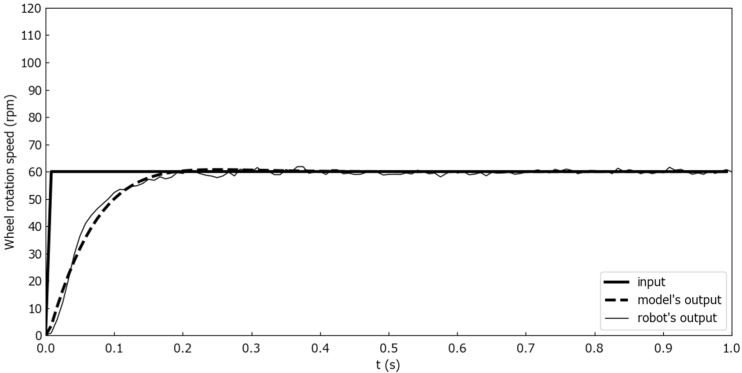
A comparison of the model outputs and the actual wheel rotation speed.

**Figure 10 sensors-25-00709-f010:**
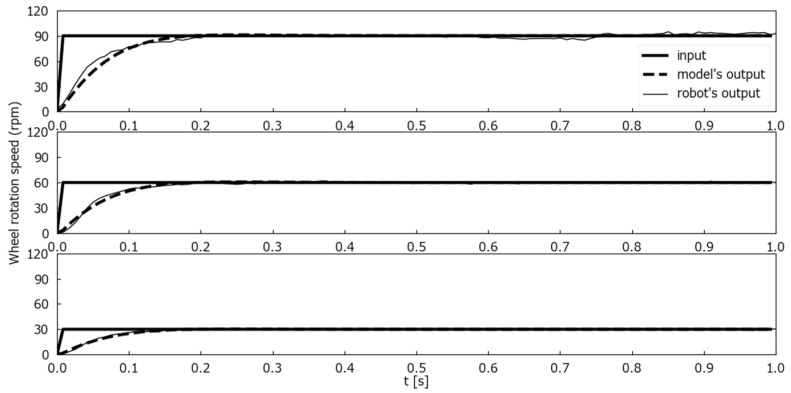
Comparison of wheel speeds between the model’s output and the actual wheel rotation speeds for three different n(t) step inputs.

**Figure 11 sensors-25-00709-f011:**
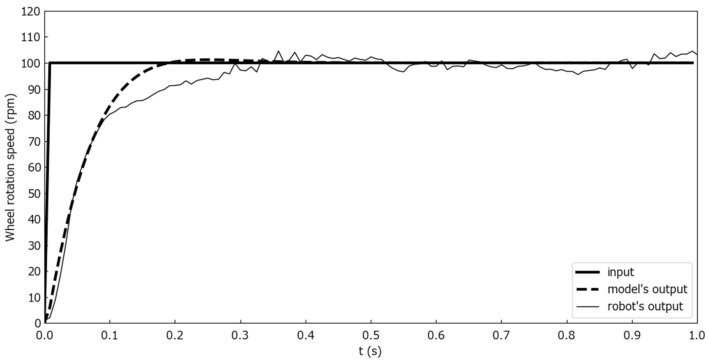
Rise delay occurring for n(t) input exceeding 100 [rpm].

**Figure 12 sensors-25-00709-f012:**
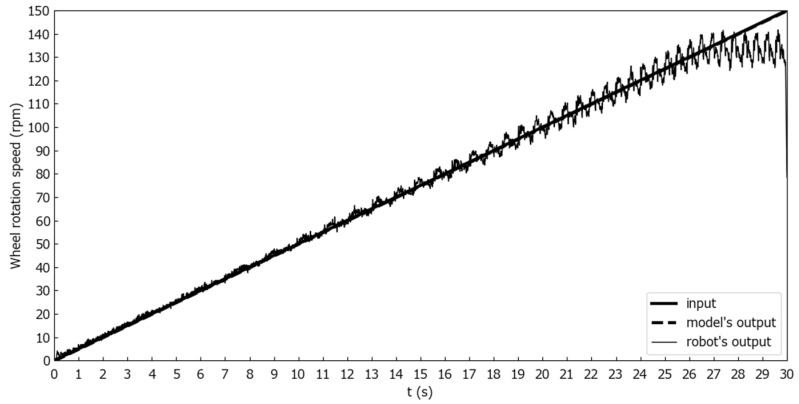
Comparison of wheel rotation speeds for ramp inputs at a slope of 5 [rpm/s].

**Figure 13 sensors-25-00709-f013:**
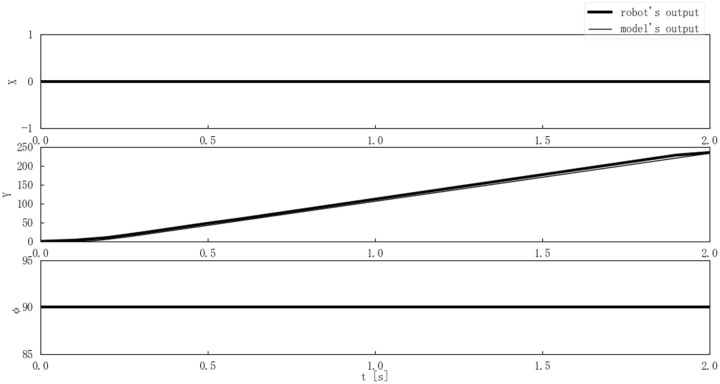
Comparison of the model and Mecanum-wheeled robot’s outputs for step inputs where n(t) is 120 [rpm].

**Figure 14 sensors-25-00709-f014:**
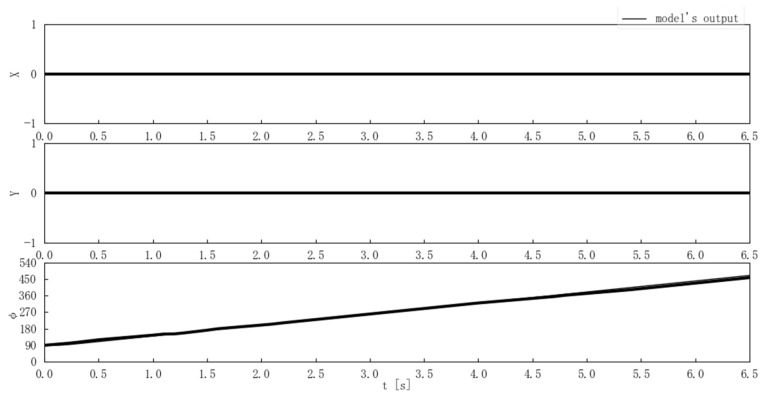
Comparison of the model and Mecanum-wheeled robot’s outputs for step inputs where ω(t) is 60 [rad/s].

**Figure 15 sensors-25-00709-f015:**
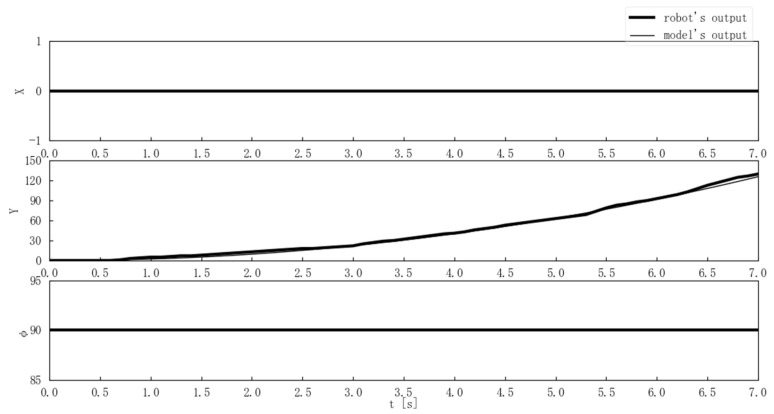
Comparison of the model and Mecanum-wheeled robot’s outputs for a ramp input with a slope of 5 [rpm/s].

**Table 1 sensors-25-00709-t001:** A comparison of the performance of each motor.

**Motor Parameters**	**BLV620KM20S-1**	**CPH62**
Rated voltage [V]	24	24
Rated power output [W]	200	170
Rated torque [N·m]	0.65	0.5
Rated rotation speed [rpm]	3000	3280
Terminal inductance [μF]	-	34.6
Terminal resistance [Ω]	-	0.31
Torque constant [Nm/A]	-	0.071
Moment of inertia [kg·m^2^]	-	1.97 × 10^−4^
Kinematic viscosity [N·m·s/rad]	-	1.20 × 10^−4^

## Data Availability

Data are contained within the article.

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
