# Peer review of "Controlling a Mecanum-Wheeled Robot with Multiple Swivel Axes Controlled by Three Commands"

_sensors, 2025, doi:10.3390/s25030709_

Round 1
Reviewer 1 Report
Comments and Suggestions for Authors
1.Why does your title emphasise the martial arts training robot? However, in your main text, you have not highlighted any particular characteristics that distinguish the Mecanum wheels of a martial arts training robot from ordinary Mecanum wheels.
2.Figure 4 should reflect how the robot's movement states correspond to the wheel motion states, rather than your current simplistic labelling.
3.The control block diagram in Figure 7 is overly simplified. It lacks detailed explanations of the controller and plant design specifications, and the feedback control loop requires more thorough description.
4.In Part Four, both the experimental procedure and results are described too briefly. There should be detailed descriptions of the output trends in each experimental graph, including explanations for any fluctuations observed. When differences are described as 'not significant', what is the threshold range for this classification?
Author Response
1.Why does your title emphasise the martial arts training robot? However, in your main text, you have not highlighted any particular characteristics that distinguish the Mecanum wheels of a martial arts training robot from ordinary Mecanum wheels.
(Ans.)
Thank you for your comment.
We have changed the title as follows.
“Control of Mecanum Wheeled Robot with Multiple Swivel Axes Controlled by 3 Commands”
2.Figure 4 should reflect how the robot's movement states correspond to the wheel motion states, rather than your current simplistic labelling.
(Ans.)
Thank you for your comment.
Arrows indicating the direction of wheel rotation have been added to Figure 4.
3.The control block diagram in Figure 7 is overly simplified. It lacks detailed explanations of the controller and plant design specifications, and the feedback control loop requires more thorough description.
(Ans.)
Thank you for your comment.
We have added Figure 7(b) as a block diagram of the plant and controller.
4.In Part Four, both the experimental procedure and results are described too briefly. There should be detailed descriptions of the output trends in each experimental graph, including explanations for any fluctuations observed. When differences are described as 'not significant', what is the threshold range for this classification?
(Ans.)
Thank you for your comment.
The following text has been added to chapter 4.
“The number of rotations output from the robot was measured with the robot's wheels not in contact with the ground. Markers were attached to the wheels, the rotation of the wheels was photographed by a camera, and the number of rotations was calculated by image processing.”
“Experimental results with step inputs as shown in Figure 10 indicate that the proposed model is able to simulate the Mecanum wheel robot's output with little error between the robot's output and the proposed model's output. As the results in Figure 11 show, there was a delay in the rise time of Mecanum wheel robot compared to the proposed model when 100 [rpm] was given as input n(t).”
“When a ramp input of 5 [rpm/s] was input as shown in Figure 12, the wheel rotation speed did not increase when the input value exceeded 130 [rpm].”
“Therefore, the hardware wheel rotation speed limit of this Mecanum wheel robot is in the neighborhood of 130 [rpm].”
“The experimental results in Figures 13-15 show that there are few errors between the Mecanum wheeled robot's output and the output of the proposed model, and the output of the proposed model is able to simulate the robot 's output.”
Reviewer 2 Report
Comments and Suggestions for Authors
This article is devoted to the problem of obtaining the model of the robot with the Mecanum wheels.
The authors derived a simplified model of the system: only the kinematics of the mecanum platform, the dynamics of the electric drives, and the models of the PI-controller of the wheel rotation speeds are taken into account. The following are not taken into account: the dynamics of the platform, which determines the cross-links between the state of the wheels; friction forces in the joints; contact effects for the rollers; effects due to the switching of the rollers (they appear, for example, in Figure 12), etc.
In the literature review, there is only one publication [8] devoted to the analysis of the influence of the above effects on the movement of omni- or Mecanum systems (contact forces are studied [8]). The review itself is carried out only on 11 sources, which is very few for the chosen topic.
The mentioned article [8] in the review is considered in the wrong context: as an article about robot control, and not about the study of contact forces. (Lines 33, 34)
Lines 78-80. To describe the motion, the parameter n(t) is introduced – the wheel rotation speed. The physical meaning of n(t) is not clearly explained: "n(t) is the wheel rotation speed when the robot moves straight forward". However, from equations (3) and (4) it follows that n(t) is the modulus of the platform center velocity, related to the wheel radius:
v_x = r*n(t)*sin(theta(t)), v_y= r*n(t)*cos(theta(t)) => |v|=sqrt(v_x^2+v_y^2)=n(t)*r
The authors obtained a simplified model of the system, describing its behavior with acceptable accuracy under certain conditions. The simplified nature of the model should be reflected in the text of the article
Author Response
In the literature review, there is only one publication [8] devoted to the analysis of the influence of the above effects on the movement of omni- or Mecanum systems (contact forces are studied [8]). The review itself is carried out only on 11 sources, which is very few for the chosen topic.
(Ans.)
Thank you for your comment.
We have added more references.
We also modified the text as follows.
“For instance, the model of Mecanum wheeled robot is built using what is completely known about the properties of the motors and motor drivers [8], study on disturbance elimination [9], study on improvement of orbit-following performance [10], study on route following control[11], study on control using SLAM [12].”
The mentioned article [8] in the review is considered in the wrong context: as an article about robot control, and not about the study of contact forces. (Lines 33, 34)
(Ans.)
Thank you for your comment.
We have modified the text as follows.
“For instance, the model of Mecanum wheeled robot is built using what is completely known about the properties of the motors and motor drivers [8], study on disturbance elimination [9], study on improvement of orbit-following performance [10], study on route following control[11], study on control using SLAM [12].”
Lines 78-80. To describe the motion, the parameter n(t) is introduced – the wheel rotation speed. The physical meaning of n(t) is not clearly explained: "n(t) is the wheel rotation speed when the robot moves straight forward". However, from equations (3) and (4) it follows that n(t) is the modulus of the platform center velocity, related to the wheel radius:
(Ans.)
Thank you for your comment.
n(t) is the parameter that determines the number of wheel revolutions specified by the robot manufacturer.
We have added the following text.
“These are the inputs defined by the robot manufacturer.”
The authors obtained a simplified model of the system, describing its behavior with acceptable accuracy under certain conditions. The simplified nature of the model should be reflected in the text of the article
(Ans.)
Thank you for your comment.
We have added the following text.
“The proposed model assumes operation in an ideal environment and models the kinematics and electrical characteristics of the robot and the controllers of the DC motors connected to the Mecanum wheels. Therefore, friction forces, contact effects among the rollers in the Mecanum wheel, and switching characteristics of the Mecanum wheel are not included in the model.”
Round 2
Reviewer 1 Report
Comments and Suggestions for Authors
1. The positive and negative angles of Mecanum wheel roller arrangement must be clearly annotated, otherwise the diagonal motion description in Figure 4 would be incorrect.
2. The final sentence of the Abstract often identifies the potential theoretical/practical implications of the study results and their relevance to the field and/or at a broader scale. Please consider modifying this sentence to better express the generalizability of the proposed approach/models.
3. Conclusions normally discuss the value provided by the current study. Please consider adding a few sentences on how the current work contributes to the existing body of knowledge on this subject to supplement this discussion of practical application value
Comments on the Quality of English LanguageThe overall quality of English language in this manuscript is good. The text is clear and comprehensible, with appropriate academic terminology and generally correct grammar usage. While there may be minor language issues that could be polished, they do not significantly impact the understanding of the technical content.
Author Response
1. The positive and negative angles of Mecanum wheel roller arrangement must be clearly annotated, otherwise the diagonal motion description in Figure 4 would be incorrect.
(Ans.)
Thank you for your comment.
We have added a figure describing the angles of Mecanum wheel roller arrangement in Figure 4.
2. The final sentence of the Abstract often identifies the potential theoretical/practical implications of the study results and their relevance to the field and/or at a broader scale. Please consider modifying this sentence to better express the generalizability of the proposed approach/models.
(Ans.)
Thank you for your comment.
We added the following sentence to the end of the abstract.
“We modeled a Mecanum Wheeled Robot using the proposed modeling method and parameter determination method, and compared the outputs of the real robot to the step and ramp inputs. The results show that the errors between the two outputs are very small and accurate enough to simulate AI learning, such as reinforcement learning, using the model of the robot.”
3. Conclusions normally discuss the value provided by the current study. Please consider adding a few sentences on how the current work contributes to the existing body of knowledge on this subject to supplement this discussion of practical application value
(Ans.)
Thank you for your comment.
We added the following sentence at the end of the Conclusions.
“This study proposed a method for determining parameters necessary for modeling a Mecanum Wheeled Robot controlled by three input commands so that the robot can be modeled even when the characteristics of the motor and motor driver are not strictly known. The proposed parameter determination method can be applied to other robots whose motor and motor driver characteristics are not strictly known. In addition, various simulations can be performed using the realized model of Mecanum Wheeled Robot. For example, the realized robot model can be used to train robot control AI for a short sword training system using a Mecanum Wheeled Robot. Therefore, this study can greatly contribute to the parameter determination method in robot modeling and to the realization of application systems using Mecanum Wheeled Robot.”